# Numerical Simulation of Morphological Changes due to the 2004 Tsunami Wave around Banda Aceh, Indonesia

**Teuku Muhammad Rasyif [1,*], Shigeru Kato [1], Syamsidik [2,3]** 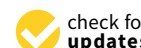 **and Takumi Okabe [1]**

[1]   Department of Architecture and Civil Engineering, Toyohashi University of Technology, 1-1 Tempaku Cho, Toyohashi, Aichi 441-8580, Japan; s-kato@ace.tut.ac.jp (S.K.); okabe@ace.tut.ac.jp (T.O.)
[2]   Tsunami and Disaster Mitigation Research Center (TDMRC), Universitas Syiah Kuala, Jl. Prof. Dr. Ibrahim Hasan, Gp. Pie, Banda Aceh 23233, Indonesia; syamsidik@tdmrc.org
[3]   Civil Engineering Department, Faculty of Engineering, Universitas Syiah Kuala, Jl. Syeh Abdurrauf No. 7, Banda Aceh 23111, Indonesia
*   Correspondence: teukumrasyif@gmail.com; Tel.: +81-80-8168-0176

**Abstract:** The 2004 Indian Ocean tsunami caused massive morphological changes around the coast of Sumatra, Indonesia. This research investigates the coastal morphological changes in the Banda Aceh area via coupling a hydrodynamic model with a sediment transport module. The Cornell Multigrid Coupled Tsunami Model (COMCOT) was coupled with the XBeach Model to simultaneously simulate sediment transport and the hydrodynamic process during the tsunami. The coupled model is known as COMCOT-SED. Field bathymetric data measured in 2006 were used to validate the coupled model. This study reveals that the tsunami's impact was more severe on the eastern part of the coast, where it hit directly. Meanwhile, the western part of the coast suffered a lower impact because of the sheltering effects from a series of small islands and a headland to the north. This study has shown that the model results from COMCOT-SED are consistent with field data and show where the tsunami waves caused offshore erosion.

**Keywords:** COMCOT-SED; tsunami; coastal erosion; sediment transport

## 1. Introduction

Extreme events, such as tsunami waves and storm surges, can cause severe change to coastal morphology. Notwithstanding the latest developments in tsunami engineering, we still do not fully understand the massive sediment transport that occurs during tsunami waves. The 2004 Indian Ocean tsunami that eroded a large part of coastal areas in affected coasts provides important evidence on tsunami sediment transport. However, large-scale simulations investigating the sediment transport process are still rare. The hydrodynamics of tsunami waves can result in massive destruction of coastal areas [1–3]. Previous research has mainly focused on morphological changes from the tsunami wave by observing the change based on satellite images, aerial photo, and field measurements [4–6]. On the other hand, very few numerical investigations of beach profile changes caused by tsunami waves have been performed [7,8].

The 2004 Indian Ocean tsunami has been studied to examine the effects of severe coastal erosion [9,10]. It detached a headland and created a small island in the case of Ujong Seudeun of Aceh [11] and Pulo Raya Island [12]. A coastal area, which had been occupied by a local community in Pulo Raya Island, was eroded by the tsunami wave. After the 2004 disaster, these areas were left unpopulated because the local community felt that returning was unsafe [12]. This shows that the

morphological changes due to the tsunami resulted in dramatic changes that affected the coastal community [12].

In recent decades, tsunami numerical simulations have been developed by applying a hydrodynamic model to simulate a tsunami wave, from its generation and propagation until inundation, such as the Cornell Multigrid Coupled Tsunami Model (COMCOT). Several benchmark studies were conducted in previous studies to evaluate the numerical model's validity with analytical solution, laboratory, and field measurements [13–15]. The COMCOT model has been used to estimate the inundation area and time of arrival known as Estimate of Inundation Area (EIA) and Estimated Time of Arrival (ETA), which contributes to the mitigation concept of creating a tsunami hazard map and incorporation with a tsunami early warning system [16]. Unlike the 2011 Great East Japan Earthquake and Tsunami that motivated a number of sediment transport studies, the effects of the 2004 Indian Ocean tsunami waves on Banda Aceh's coastal morphology have not been fully analyzed despite the severe damage. So far, the coastal morphological changes in Banda Aceh have been analyzed by observing the change from the satellite imagery and measurement data [17,18].

To help uncover the effects of the 2004 Indian Ocean tsunami, we obtained bathymetric data, collected by Indonesian Public Works right after the 2004 tsunami in Banda Aceh. The data helped us numerically investigate the tsunami's effect on the Banda Aceh coasts. We used a modified COMCOT model, coupled with a sediment module (COMCOT-SED), to integrate hydrodynamic and sediment transport simulations in one simultaneous process. This model was used to investigate the tsunami wave's impact on morphological changes in Lhok Nga and Thailand [19,20]. The studies revealed that the COMCOT-SED model results were consistent with measurements of tsunami deposit data. This model has also been used to investigate the impact of morphological change from a future tsunami in Painan, West Sumatra, Indonesia [21].

This paper is aimed at investigating the impacts of the 2004 Indian Ocean tsunami on coastal morphological changes at Banda Aceh. One set of bathymetric data that was measured before and after the 2004 tsunami and geospatial analysis data were used for validation. The study was performed to better understand the sedimentation-erosion process driven by tsunami waves. The results of this study are expected to contribute to tsunami mitigation for coastal areas based on the real event in Banda Aceh city.

This paper is organized as follows. We begin by explaining the study area's characteristics before and after the 2004 tsunami. In the next sub-section, we elucidate study methods and how the validation process was performed. Finally, we discuss our findings and further potential uses of this study before presenting conclusions.

## 2. Study Area

Banda Aceh, which is located on the northern end of Sumatra Island, is the capital city of the Aceh Province of Indonesia (Figure 1). Banda Aceh is bordered by the Malaccaa strait, the Aceh Besar district, and the Indian Ocean on the north, south, and west, respectively. Banda Aceh is vulnerable to tsunami disaster because of its geographic location, which is close to the Sumatran subduction zone. Banda Aceh is characterized by relatively flat topography which can cause the tsunami waves to inundate areas further inland, as was the case during the 2004 Indian Ocean tsunami [22]. Approximately 67 tsunami poles have been built in Banda Aceh for disaster education, as a tsunami memorial, and an evacuation sign by the local people and the Embassy of Japan in Indonesia through the Umi Abasiah Foundation. Each tsunami memorial pole illustrates the maximum tsunami wave height at that location based on local witnesses and watermarks. We used these data points to compare with our model results.

Banda Aceh is divided into eight sub-districts, four of which share boundaries with that coastal area, namely Meuraxa, Kuta Raja, Kuta Alam, and Syiah Kuala. Banda Aceh borders with Aceh Besar district, as indicated by white color in Figure 1b.The total Banda Aceh area is approximately 61.36 km$^2$, as indicated by orange color in Figure 1a. The city has a coastline of approximately 12 km.

The pre-tsunami Banda Aceh coastal condition is indicated by gray and green colors in Figure 1b. After the tsunami event, the Banda Aceh coastal area underwent massive morphological changes within the area shown in green. The orange color in Figure 1 shows buildings within the Banda Aceh city. Due to the lack of knowledge about tsunami disasters, there was little land use management information for Banda Aceh before 2004.

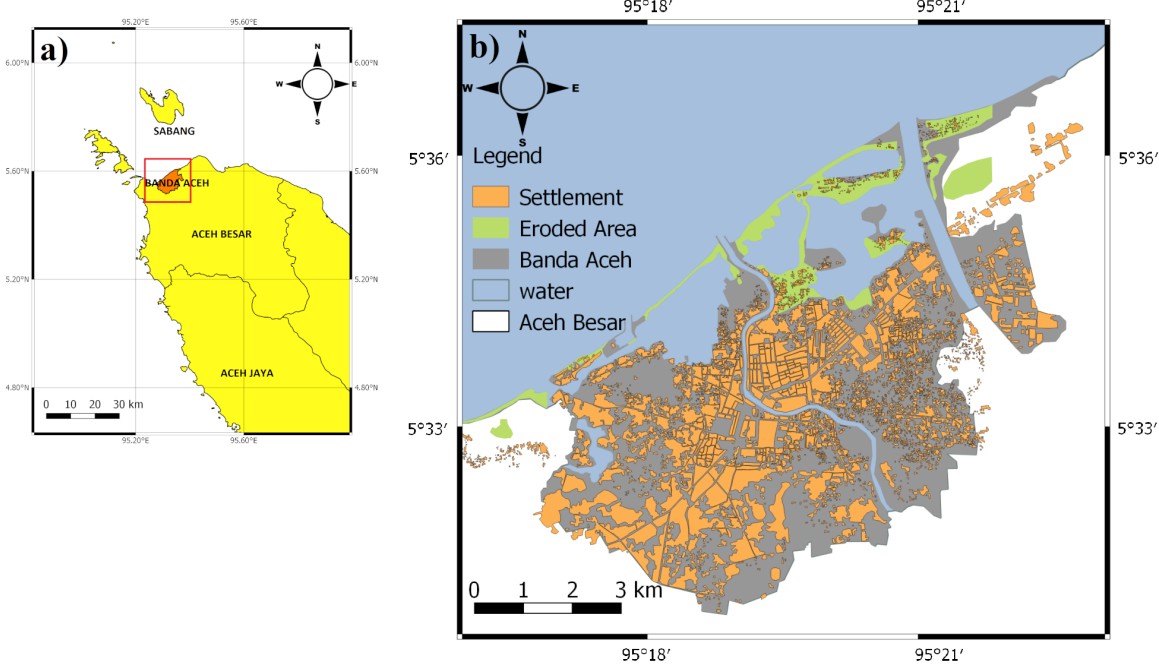

**Figure 1.** (**a**) Study area and (**b**) Banda Aceh city before and after the 2004 Indian Ocean Tsunami (green color indicates the land area eroded by the tsunami). This figure was modified from [23].

## 3. Methodology

### 3.1. Hydrodynamic Model

The morphological changes that occurred in Banda Aceh during the 2004 Indian Ocean Tsunami were simulated using the COMCOT-SED model. It is an open source model developed by Dr. Li's group at Nanyang Technological University. The model itself consists of two open source packages: the COMCOT and the Xbeach. The COMCOT-SED model's performance had been validated to investigate the tsunami-induced morphological changes in Lhok Nga, Aceh and Thailand, but not in Banda Aceh [19,20]. The model could simulate morphological changes and was consistent with sediment deposition measurements.

A set of linear and non-linear shallow water equations (SWEs) was applied to the COMCOT-SED model to calculate the water surface elevation and volume flux. Both equations are available with spherical or Cartesian coordinate systems. In the linear SWEs with spherical coordinates in a large area, the Coriolis effect due to the earth's rotation is included. In the local simulation area, the non-linear SWEs with the Cartesian coordinate system is adopted. The Cartesian coordinates, with non-linear SWEs on the innermost layer, must be used to apply the sediment calculation in the COMCOT-SED model. Equations (1)–(3) show conservation of mass and momentum applied in the Cartesian coordinate system (details of the equations can be seen in the COMCOT manual [15]). In Equations (1)–(3), $\eta$ represents the water surface elevation; $h$ is the water depth; $H = \eta + h$ is the

total water depth; $u$ and $v$ are depth averaged velocities in the $x$ and $y$ direction, respectively; $g$ is the gravitational acceleration; $n$ is Manning's coefficient for bottom roughness.

$$\frac{\partial \eta}{\partial t} + \frac{\partial (Hu)}{\partial x} + \frac{\partial (Hv)}{\partial y} = 0 \tag{1}$$

$$\frac{\partial u}{\partial t} + u\frac{\partial u}{\partial x} + v\frac{\partial u}{\partial y} + \frac{gn^2}{H^{4/3}} u \sqrt{u^2 + v^2} = -g\frac{\partial \eta}{\partial x} \tag{2}$$

$$\frac{\partial v}{\partial t} + u\frac{\partial v}{\partial x} + v\frac{\partial v}{\partial y} + \frac{gn^2}{H^{4/3}} v \sqrt{u^2 + v^2} = -g\frac{\partial \eta}{\partial y} \tag{3}$$

### 3.2. Sediment Model

The sediment calculation is based on the XBeach Model version 6 [24]. Xbeach is applied to a depth-averaged advection–diffusion equation according to [25] to calculate a sediment concentration as shown in Equation (4).

$$\frac{\partial HC}{\partial t} + \frac{\partial HCu}{\partial x} + \frac{\partial HCv}{\partial y} + \frac{\partial}{\partial x}\left(D_h H \frac{\partial C}{\partial x}\right) + \frac{\partial}{\partial y}\left(D_h H \frac{\partial C}{\partial y}\right) = \frac{HC_{eq} - HC}{T_s} \tag{4}$$

where $C$ represents the depth-averaged concentration of suspended sediment; $D_h$ is the sediment diffusion coefficient. A default value of $D_h = 1.0$ was selected. $T_s$ is the adaptation time of sediment concentration that is calculated based on total water depth ($H$) and sediment fall velocity ($w_s$) as in the following equation:

$$T_s = max\left(f_{Ts}\frac{H}{w_s}, T_{s,min}\right) \tag{5}$$

where $f_{Ts}$ is a correction and calibration factor. $f_{Ts}$ has a range and a default value of about 0.01–1 and 0.1, respectively. The default value of $f_{Ts} = 0.1$ was adopted in the simulation. $T_{s,min}$ is the minimum adaptation time. $T_{s,min}$ has a range and a default value of about 0.01–10 and 0.5, respectively. $T_{s,min} = 0.2$ was adopted in the simulation. These parameters were applied following previous studies [20,26].

$C_{eq}$ represents the equilibrium sediment concentration calculated according to [21] as the following approximation:

$$C_{eq} = \frac{q_s + q_b}{|u|H} \tag{6}$$

where $q_s$ is the sediment volume flux for the suspended load, $q_b$ is the sediment volume flux for the bed load, and $|u|$ is the magnitude of the depth-average velocity. The volume fluxes for the suspended and bed loads were calculated based on [27] as the following expressions:

$$q_s = 0.012|u|H\left(\frac{|u| - u_{cr}}{[(s-1)\,gd_{50}]^{1/2}}\right)^{2.4}\left(\frac{d_{50}}{H}\right)D_*^{-0.6} \tag{7}$$

$$q_b = 0.005|u|H\left(\frac{|u| - u_{cr}}{[(s-1)\,gd_{50}]^{1/2}}\right)^{2.4}\left(\frac{d_{50}}{H}\right)^{1.2} \tag{8}$$

where $D_* = [(s-1)g/v^2]^{1/3}d_{50}$ is the dimensionless particle diameter. $v$ is kinematic viscosity. $d_{50}$ is the median grain size. $u_{cr}$ is the critical depth-averaged flow velocity that was calculated according to [27] as the following:

$$u_{cr} = 0.19\,(d_{50})^{0.1}\log\left(12H/3d_{90}\right) \tag{9}$$

when 0.0001(m) $\leq d_{50} \leq$ 0.0005(m) and

$$u_{cr} = 8.5\,(d_{50})^{0.6}\log\left(12H/3d_{90}\right) \tag{10}$$

when $0.0005(\text{m}) \leq d_{50} \leq 0.002(\text{m})$. $d_{90} = 1.5 d_{50}$ represents the diameter where 90% of the distribution has a smaller particle size.

*3.3. Bottom Change Model*

Equation (11) shows the mass-balance for bed-level changes used to update the bottom elevation. This can be expressed in the form of a partial differential equation describing the process of the sediment transport rate in the $x$ and $y$ directions, respectively. $z_b$ represents the bottom elevation that changes with time. $p$ is the bed material porosity. $q_x$ and $q_y$ represent the sediment transport rates in the $x$ and $y$ directions, respectively [24].

$$\frac{\partial z_b}{\partial t} + \frac{1}{1-p}\left(\frac{\partial q_x}{\partial x} + \frac{\partial q_y}{\partial y}\right) = 0 \tag{11}$$

$$q_x = \frac{\partial HCu}{\partial x} + \frac{\partial}{\partial x}\left[D_h H \frac{\partial C}{\partial x}\right] \tag{12}$$

$$q_y = \frac{\partial HCv}{\partial y} + \frac{\partial}{\partial y}\left[D_h H \frac{\partial C}{\partial y}\right] \tag{13}$$

## 4. Model Setup

*4.1. Initial Condition*

Several researchers proposed a multi-fault model to represent the 2004 Indian Ocean earthquake because of its complexity. The earthquake's rupture area extended approximately 1200 km from the epicenter, in western Aceh Province, to the Andaman Islands. The fault parameters published by [28] were applied in this study to generate the bottom deformation as an initial trigger for the tsunami, as shown in Figure 2. The source was divided into five segments with a total seismic moment equivalent to $M_w = 9.22$. The fault parameters were validated by comparing the simulation results to satellite transect, tide gauge, and run-up height measurements near the Banda Aceh coast [28]. The bottom deformation was generated using the deformation model proposed according to [29]. The COMCOT-SED model assumed that the sea surface changes instantaneously followed the deformation of the sea bed as the initial condition for the tsunami wave in 2004. This is the same model that was used to investigate the morphological changes to the Khao Lak coast in Thailand during the 2004 tsunami [20].

*4.2. Grid Setup and Input Data*

The multi-layer system in the COMCOT-SED model was applied in this study to obtain accurate and detailed results for the specific simulation area. As a result, the morphological changes in Banda Aceh can be observed based on high-resolution data. Five nested grid layers were used to simulate the tsunami wave from the earthquake source to Banda Aceh city, as shown in Figure 3. In layers 1 to 4, only the hydrodynamic model was applied. Both the hydrodynamic and sediment transport models were applied in layer 5. The time step with about 0.01 s was adopted in all layers for calculation stability. Table 1 shows the input parameters implemented on every layer in this study, such as grid size, simulation area, coordinate system, and SWE type.

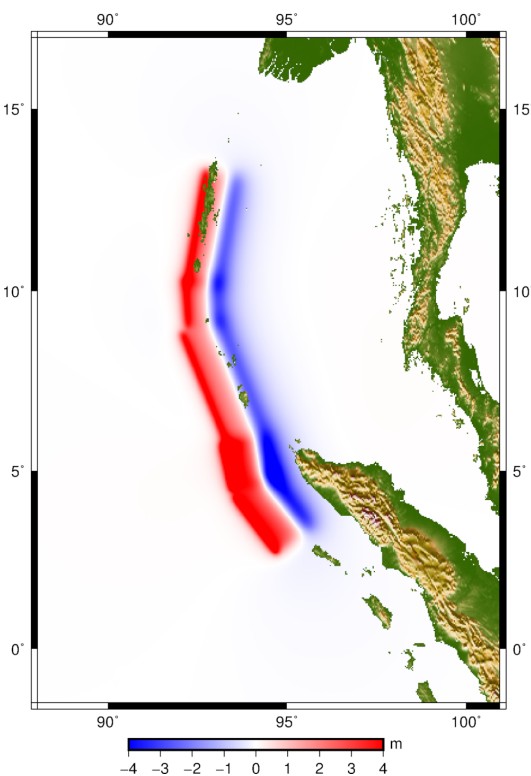

**Figure 2.** The initial condition (water surface elevation) for the 2004 Indian Ocean Earthquake.

**Table 1.** Information on the setup of the five layers for simulation.

|  | Layer 01 | Layer 02 | Layer 03 | Layer 04 | Layer 05 |
|---|---|---|---|---|---|
| Number of grids | 780 × 900 | 1080 × 900 | 810 × 783 | 567 × 405 | 793 × 1032 |
| Lati. (degree) | 88–101 | 92–98 | 94.5–96 | 95.15–95.5 | 95.275–95.386 |
| Longi. (degree) | 0–15 | 4–9 | 5.25–6.7 | 5.5–5.75 | 5.507–5.646 |
| Grid size (m) | 1840 | 613.33 | 204.4 | 68.148 | 17.037 |
| Grid size ratio |  | 3 | 3 | 3 | 4 |
| Coordinate system | Spherical | Spherical | Spherical | Cartesian | Cartesian |
| SWE | Linear | Linear | Linear | Non-linear | Non-linear |

General Bathymetric Chart of the Ocean (GEBCO) data [30] were adopted in this study for the bathymetry and topography data for layers 1–4. The GEBCO data are open source for general bathymetry and topography data with a resolution of about 0.5 arcminute. Layers 1–4 have been implemented for different grid sizes, as shown in Table 1, that are smaller than the GEBCO data. Therefore, GMT (Generic Mapping Tools) [31] software was used to interpolate the data for layers 1–4. The topography data from GEBCO were selected despite having low accuracy, because it was impossible to get the data just before the 2004 tsunami for this study. The bathymetry for the innermost layer (layer 5) was developed using nautical chart data measured by Dishidros of TNI AL Indonesia in 2001 with a map scale of approximately 1:100,000. The nautical chart map was digitized, using the geo-referenced function in Quantum GIS software [32]. Quickin and Refgrid software in Delft3D open source [33] was used to combine the digitized bathymetry data from the nautical chart with a topography data from GEBCO in Layer 5.

The sediment data were obtained from TDMRC conducted at Ulee Lheu bay in 2006. The grain size $d_{50}$ was specified as 0.18 mm for the sea area. The land area was assumed to have a similar grain size to the sea area. The entire area simulation for layer 5 was considered as erodible area.

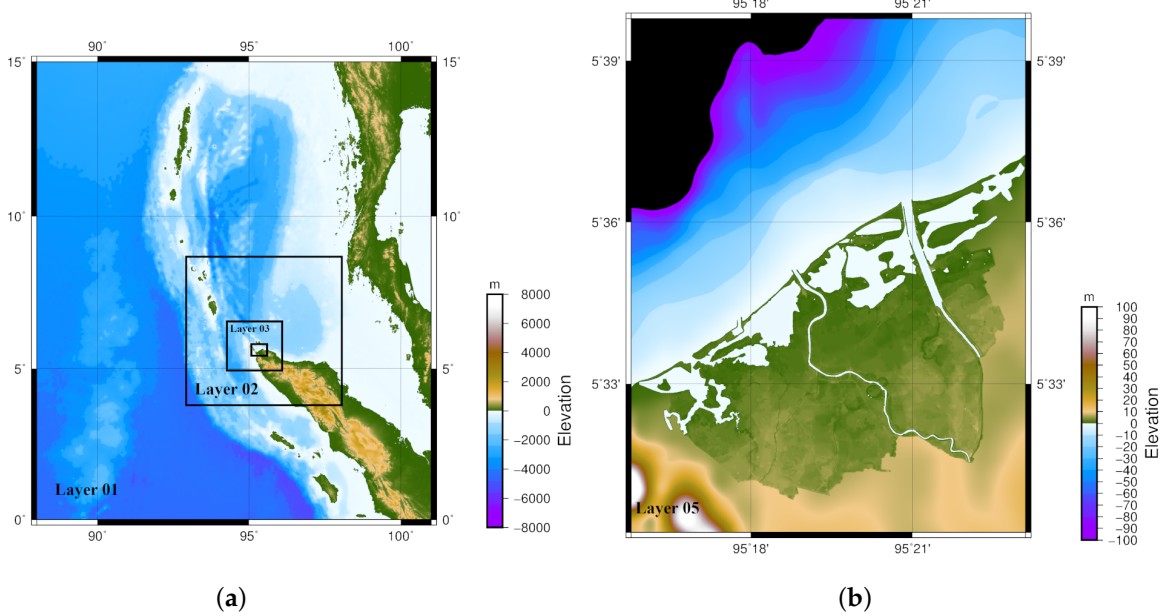

(**a**)          (**b**)

**Figure 3.** Nested grids for the Cornell Multigrid Coupled Tsunami Model coupled with a sediment module (COMCOT-SED) simulation domain showing (**a**) Layers 1–4 and (**b**) Layer 5.

Figure 4 shows the distribution of the Manning coefficients used in layer 5 as the innermost layer. A satellite image, based on Google Earth data in 2004, was applied to this study to show land use in Banda Aceh city. Different Manning coefficient values were obtained based on the land use classification in Banda Aceh city before the 2004 Indian Ocean tsunami. The Manning's values were adopted from previous studies by [19,34], as shown in the Figure 4 legend. Settlements comprised most of the land area in Banda Aceh city. Ponds and mangrove areas existed on the backside of the coastal area, as indicated by blue and green colors. A uniform Manning coefficient value of 0.013 was used for layers 1–4.

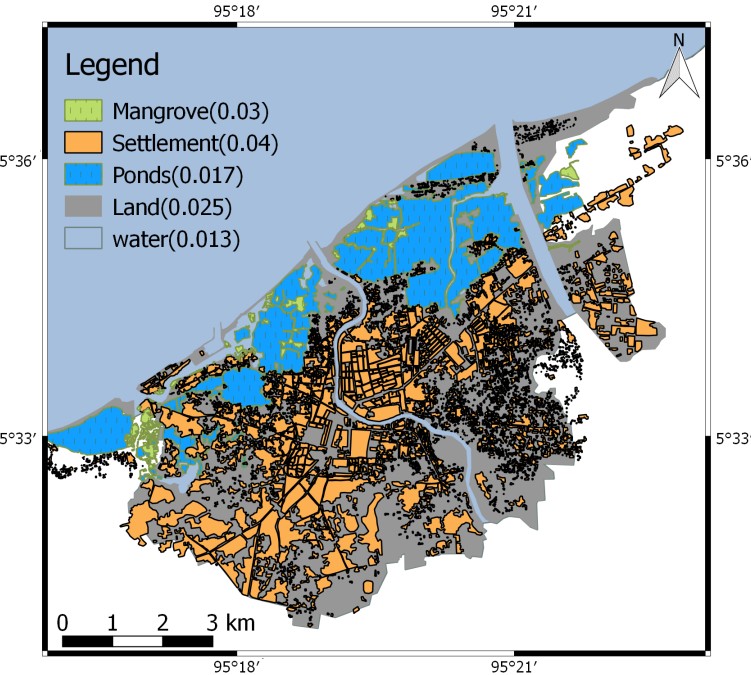

**Figure 4.** Distribution of Manning Coefficients in COMCOT-SED simulation.

### 4.3. Observation Points and Cross Section

Several observation points and cross-sections were placed on the simulation area to collect information such as water level and bed level on layer 5, as shown in Figure 5. The observation points (marked by red dots) were the same as the tsunami pole locations according to [35]. These locations were used to compare the maximum tsunami water level between the simulation results and the actual data. Three cross-sections perpendicular to the coastline were used to compare the coastal profile before and after the tsunami. The cross-sections were chosen because the tsunami wave caused massive damage at these locations.

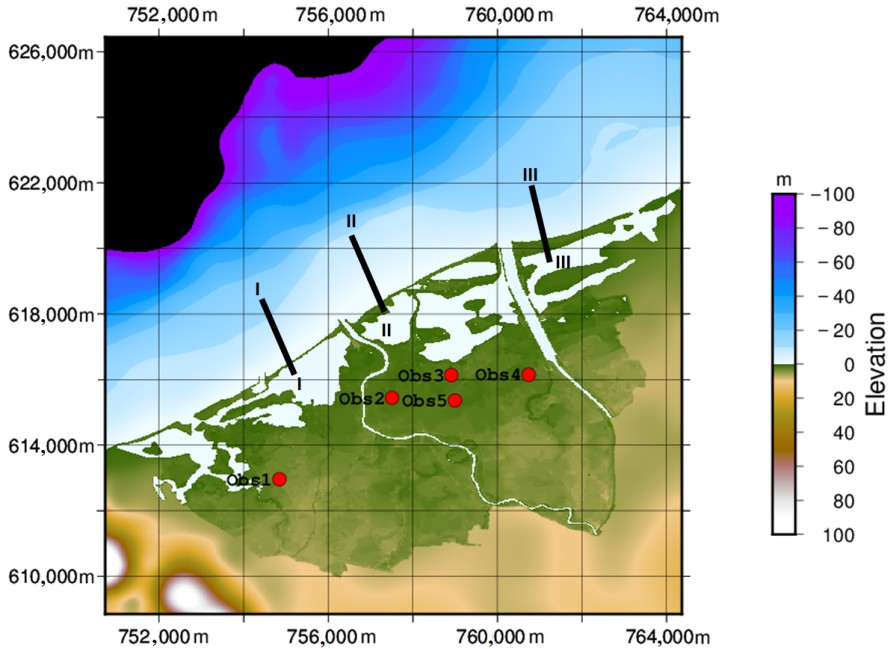

**Figure 5.** Locations of the numerical observation points and cross-sections.

## 5. Results and Analysis

### 5.1. Inundation Map

Banda Aceh city was inundated by the tsunami wave with the maximum inundation depth of approximately 7 m at the front of coastal land barrier and 4 m in the city area based on the simulation results, respectively (Figure 6). The tsunami wave flooded toward Banda Aceh city with a maximum inundation distance of about 4 km from the shoreline according to field observation [22]. The simulation results predicted a distance of about 5.5 km from the shoreline, which overestimated the inundation of the real event. This can be explained because Banda Aceh city had a lot of structures around the coastal area which were not reproduced in the simulation, and they can reduce the tsunami wave energy and limit inundation further inland.

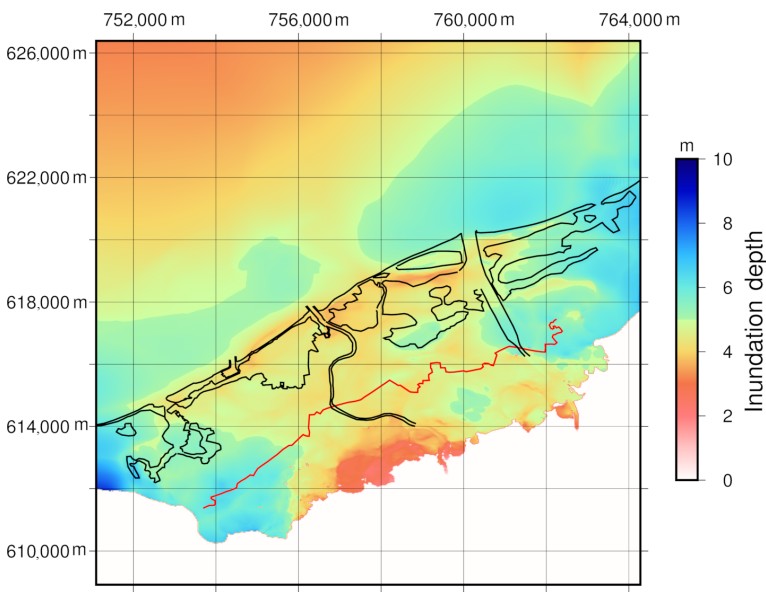

**Figure 6.** Map of the maximum inundation depth (m). The red line indicates the inundation limit based on the real event. The black line marks the Banda Aceh shoreline before the tsunami.

Figure 7 compares the simulated maximum tsunami height and measured height at the tsunami poles indicated in Figure 5. The simulated results at all points overestimate the measured results. However, the differences between the simulated result and the measured one at obs2, 3 and 5 are not so large. These points are located in the middle of Banda Aceh city. Therefore, the tsunami inundation in the city area was simulated appropriately. On the contrary, a large difference is demonstrated at obs4. The location of obs4 is close to a large river. In Figure 6 of the spatial distribution of the maximum inundation depth, a large inundation depth area, in blue color, extends further in the inland area than the area close to the river mouth. Based on these results, it is suggested that the tsunami propagation in the river occurred up to the inland area and the significant overestimate of the maximum height was caused by the flood from the river in the simulation. This result also indicated the risk of river flooding caused by the tsunami in the inland area.

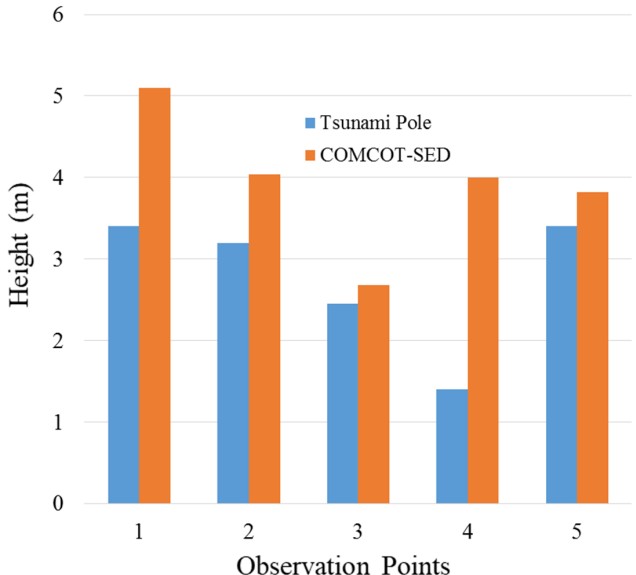

**Figure 7.** The calculated tsunami heights (orange bar) and surveyed data (blue bar) in Banda Aceh city.

### 5.2. Sedimentation and Erosion

Figure 8 shows a map of erosion and deposition thickness by tsunami inundation 90 min after the tsunami (earthquake) occurrence according to the COMCOT-SED model. The results predict that the tsunami wave caused morphological changes along the entire Banda Aceh coast; especially severe erosion is shown in the red color areas. The most eroded area is along the eastern part of the coast of Banda Aceh because of direct attack of the tsunami waves. The western part of the coast of Banda Aceh was relatively shielded by a headland and a group of small islands as shown in Figure 1a. The sediment eroded by the tsunami wave near the coast line was transported to the land side and deposited in the ponds and lagoons behind the coast, as indicated by blue color in Figure 4. Less erosion and deposition occurred in coastal areas behind the coastal land barrier and pool areas.

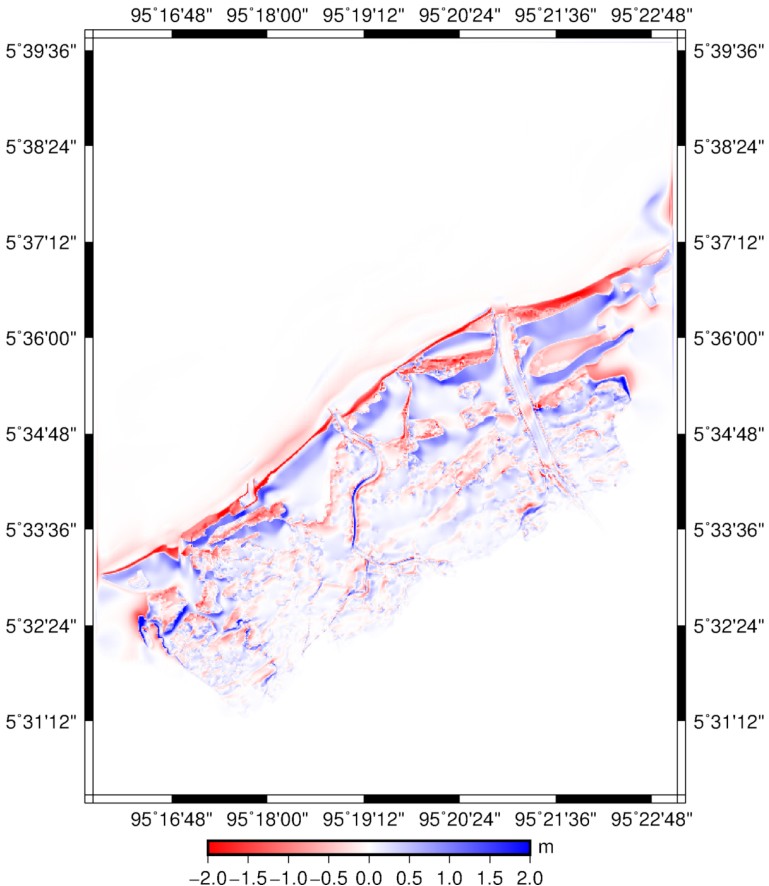

**Figure 8.** Map of erosion and deposition areas. Red color indicates negative (erosion) and blue indicates positive (deposition).

### 5.3. Coastal Profile Changes

Figure 9 shows sea bed profiles for the initial condition, the simulated result by COMCOT-SED, and measured data at cross-sections I-I, II-II, and III-III in Figure 5. The post-tsunami profile measurement was conducted in 2006 by the Public Works Department of Aceh two years after the tsunami event. In the initial condition, the shoreline position was at a distance of 600, 270, and 500 m from the ends of sections I-I, II-II, and III-III, respectively. In the simulated profiles, the coastal area directly facing the sea was eroded, reducing the height. In contrast, deposition occurred in the fishponds, as indicated by the black arrow in Figure 9a. Along this coast, the land barriers were eroded by the tsunami wave, and bed profiles in very shallow areas became flat at the distance from 0 to 300 m and from 0 to 550 m for sections II-II and III-III, respectively.

Comparing the simulation results to the 2006 bathymetry data indicates that the bathymetry profile underwent further changes. The disagreements can be caused possibly by other morphological changes

in the 2004–2006 time range, low accuracy of the topography data before the tsunami, the uniform grain size applied for the simulation, and limitations of the current model. The low accuracy of topography data and limited information of grain size can influence the location of erosion and deposition in numerical calculation [36]. The current model also has limitations in calculating the bed stress based on the Manning formula that possibly causes overestimation. Coupling the SWE and a Reynolds-averaged Navier–Stokes (RANS) model with a boundary-layer method indicates that the boundary-layer method is able to calculate the bed stress accurately compared to the Manning formula [37,38]. The advantage of the boundary-layer method was applied by [39] to predict the tsunami wave-induced scouring. This suggests that it is important to apply the boundary-layer method in the model to calculate the sediment transport properly. In general, coastal erosion was indicated by the 2006 bathymetry data, as shown in Figure 9b,c with distances of 0–300 m and 50–600 m. This is shown by the simulation results at the cross-sections II and III, where the land barrier had been destroyed by the tsunami wave.

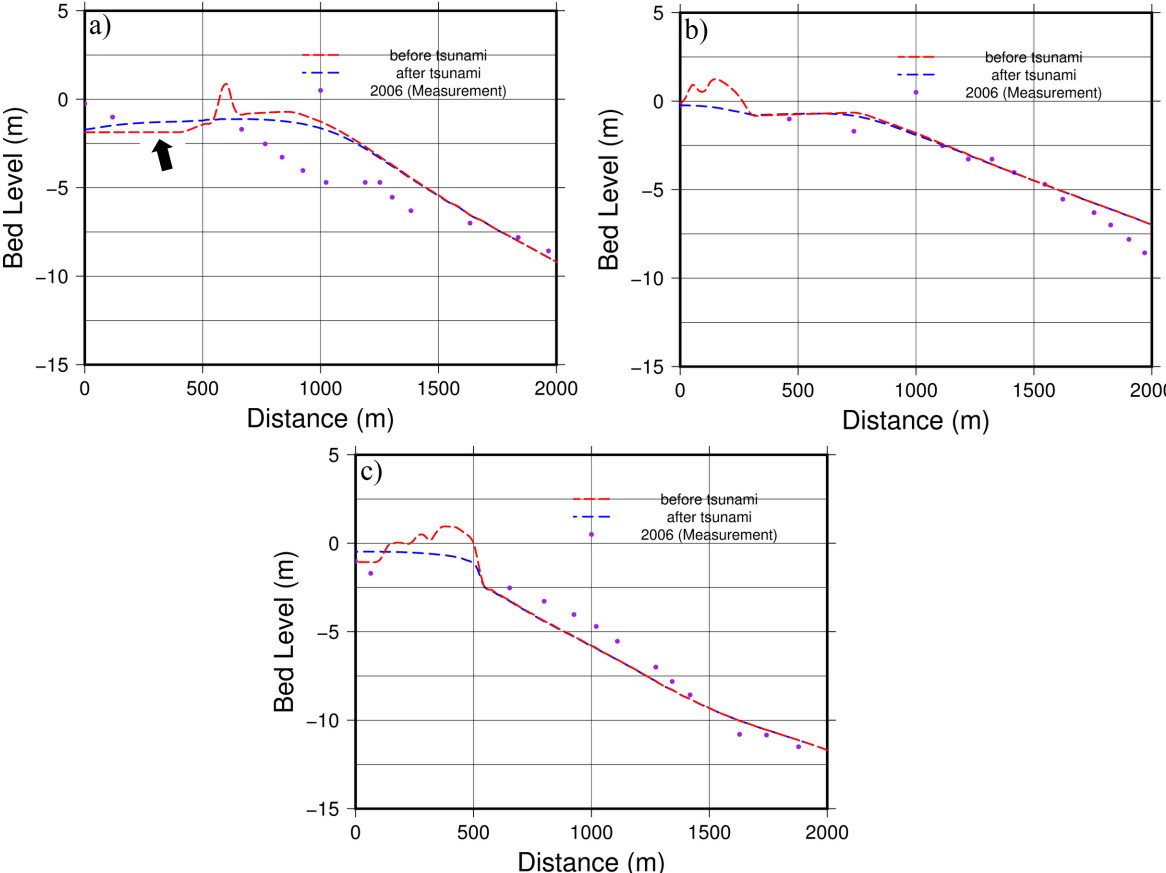

**Figure 9.** Bed profile for the initial condition (indicated by red line), after tsunami inundation (simulation) by the 2004 Indian Ocean Tsunami (indicated by blue line), and measured in 2006 (indicated by pink-solid circles) in the cross-section; (**a**) I-I, (**b**) II-II and (**c**) III-III.

## 5.4. Destroyed Land Barrier

This part of the study used geospatial information, based on a Google Earth map released between 2004 and 2005, to validate the simulation result. Figure 10a,b displays the Banda Aceh coast condition in 2004, (a) before the tsunami and (b) afterwards. The topographic data in the simulation results are also shown in Figure 10c,d. Figure 10c is the initial topography (before the tsunami) and Figure 10d is the simulated topography after the tsunami impact. Figure 10a,c, and 10b,d should be compared. The northern coast area underwent massive morphological changes after the tsunami. The beach area from lines A1 to A2 and C1 to C2 in Figure 10d was completely eroded by the tsunami wave. This area was natural beach without any protection; therefore, the tsunami wave could erode easily in this area.

The beach area from line B1 to line B2 shows differences between the satellite image and the simulation results. This could be because no structures were assumed and the whole area was consider erodible in the simulation but there were several hard structures on the ground before the tsunami, especially the area indicated by a black arrow shown in Figure 10a. At this area, a port structure complex prevented severe erosion from the tsunami waves.

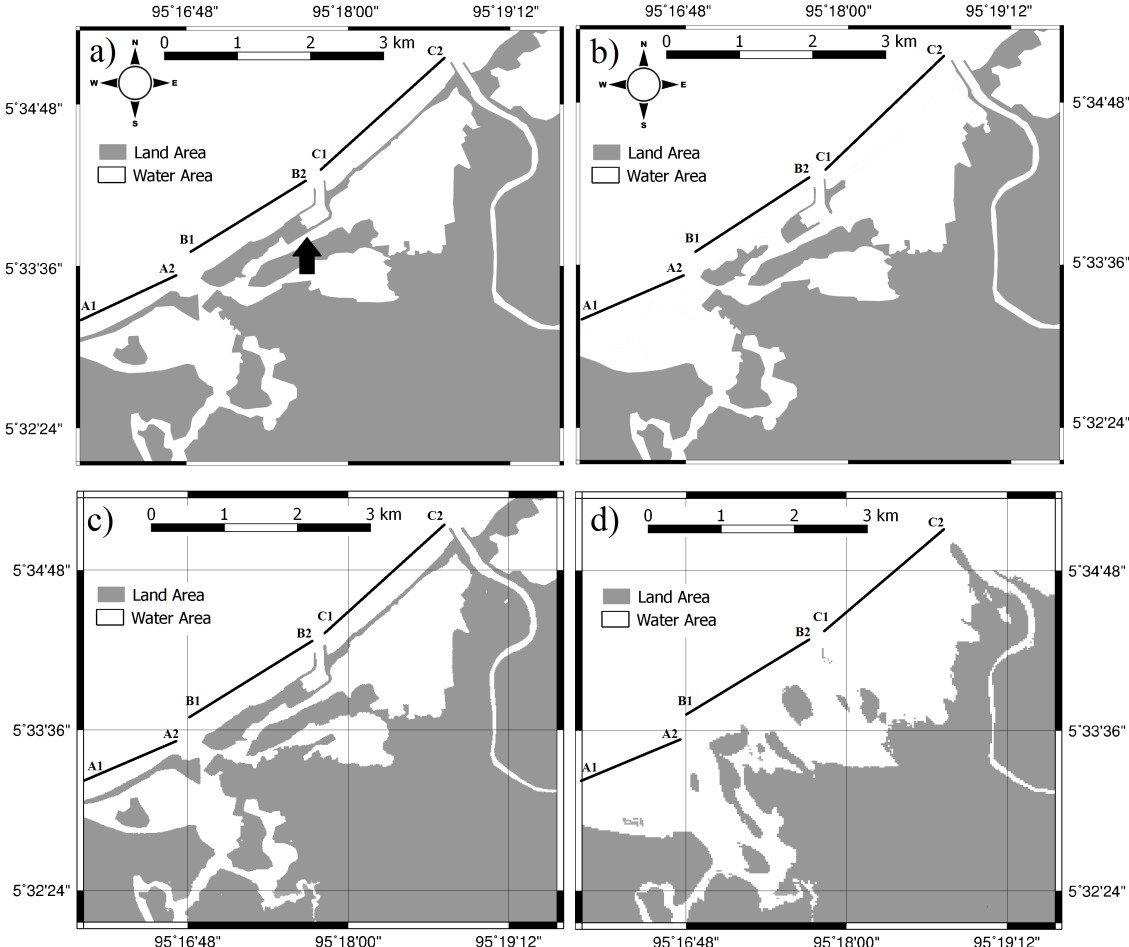

**Figure 10.** Coastal area comparison between the satellite images: (**a**) before, (**b**) after the tsunami in 2004 and numerical simulation: (**c**) before simulated and (**d**) after simulated by COMCOT-SED. The satellite images are taken from Google Earth with time history imagery setting on 23 July 2004 and 28 January 2005.

Geospatial analysis results based on [23] were used in this study to compare the simulation results, as shown in Figure 11a. The coastal area in Syiah Kuala sub-district, indicated from point E1 to E2 as shown in Figure 11c, was destroyed by the tsunami wave based on the simulation results. The simulation results indicated a similar result to the geospatial analysis, indicating that this area was also massively impacted by the tsunami waves. This area was natural beach without any protection and was used as a settlement area before the 2004 tsunami. The coastal area was completely eroded during the tsunami event in 2004.

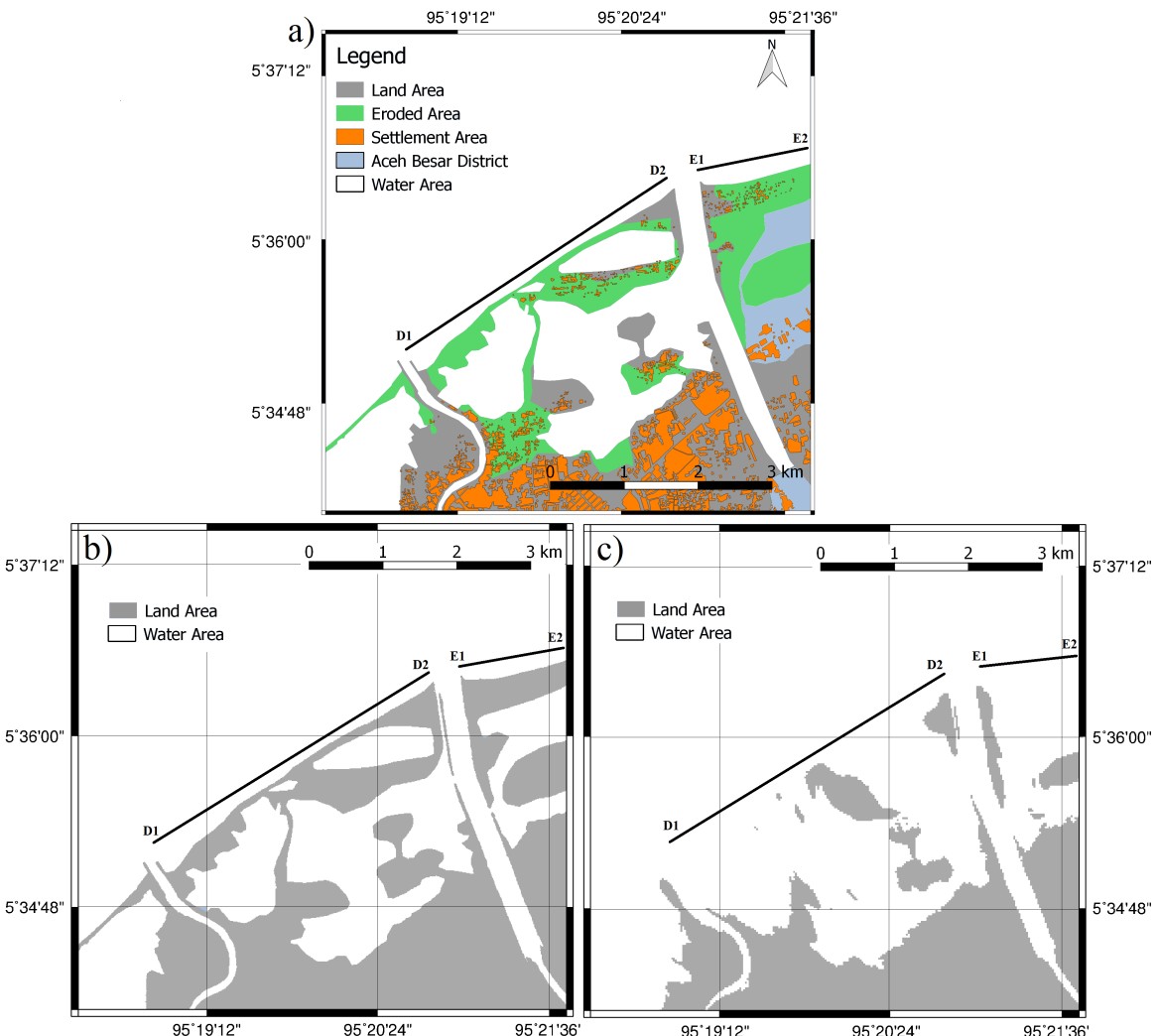

**Figure 11.** Coastal area comparison between the geospatial data and numerical simulation. (**a**) Green and a gray areas indicate the land condition before the tsunami in 2004. The green area indicates the eroded area after the tsunami in 2004. This figure was modified from [23]. (**b**) before simulated and (**c**) after simulated by COMCOT-SED.

The coastal area from point D1 to D2, as shown in Figure 11, is located in Kuta Alam and Syiah Kuala district. These areas were fish/shrimp ponds and vegetation before the 2004 tsunami. A small, but dense, housing area also existed here before the tsunami. The simulation results indicated that this area underwent massive morphological changes due to the tsunami wave (Figure 11c). The land area, which had been used as ponds, vegetation, and a settlement area, was covered with water [23]. The coastal area located behind the ponds did not receive massive morphological changes. The simulation results are consistent with the geospatial analysis in which the land barrier had been destroyed.

## 6. Conclusions

This study was conducted to analyze the 2004 tsunami wave's impact on the Banda Aceh coastal area. The COMCOT-SED model was used to simulate morphological changes caused by the tsunami wave. The simulation results were validated by comparing the sedimentation and erosion areas between the simulation results and geospatial information based on Google Earth map data from 2004 to 2005 and Syamsidik et al. [23]. This study also compared the simulation results to bathymetry data collected in 2006. The simulation results show that the tsunami wave caused massive

morphological changes to the Banda Aceh coastal area. The land barrier area, which was in a natural condition, was almost completely eroded by the tsunami wave. The simulation results agreed well with geospatial data.

Numerical simulation can be used to investigate morphological changes due to tsunami waves in the high-risk area. This model can predict areas of sedimentation and erosion caused by a tsunami wave. This study shows that the numerical model can simulate destruction of the land barrier by the 2004 tsunami. Unfortunately, the land barrier on Banda Aceh coast was used by the local community as a settlement. The 2004 tsunami wave destroyed not only the land barrier but also the settlement area in Banda Aceh. The information of erosion and sedimentation location that obtained from simulation results can be used by local governments for land use management to help avoid a disaster such as the one in Banda Aceh during the 2004 Indian Ocean tsunami.

This study has certain limitations. The low accuracy of the data can cause differences between the simulation results and field observation. The input data, especially the good accuracy of topography and bathymetry data in the simulation, can influence the inundation extent in the simulation results as indicated by previous researches [40,41]. The other study also agree that the low accuracy of the data, such as topography, bathymetry, and sediment grain size distribution, can influence the erosion and deposition area in the sediment transport model [36]. Further study is needed to investigate the morphological changes caused by future tsunamis, based on the current condition of Banda Aceh city, to enhance tsunami disaster mitigation by analyzing the sensitivity of the input data that are applied in the sediment transport model.

**Author Contributions:** T.M.R. conducted the numerical simulation and compared the simulated and measured results. S. provided the measured data at cross-sections. T.M.R. wrote the manuscript with contribution from the other co-author. S.K. and O.T. supervised the work. S.K and S. reviewed and edited the manuscript. All authors contributed to the discussion and interpretation of the results.

**Funding:** This study has been supported by the Ministry of Education, Culture, Sports, Science and Technology of Japan (MEXT scholarship). Authors are also grateful to Ministry of Research, Technology, and Higher Education of Indonesia (RISTEKDIKTI) for a research grant under International Collaboration and Publication (PKLN) 2018 that has enable Shigeru Kato and Syamsidik to collaborate, Contract No. SK.60/UN11.2/SP3/2018.

**Acknowledgments:** Author would like to thank Linlin Li from Earth Observatory of Singapore for providing the COMCOT-SED model for this study. Also, we thank to Tursina Musa and Inayah Zhiaul Fajri of researchers in TMDRC of Syiah Kuala University for the bathymetry data and land use data for Banda Aceh city in 2006 and before tsunami in 2004, respectively.

**Conflicts of Interest:** The authors declare no conflict of interest.

## Abbreviations

The following abbreviations are used in this manuscript:

| | |
|---|---|
| COMCOT | Cornell Multigrid Coupled Tsunami Model |
| EIA | Estimated of Inundation Area |
| ETA | Estimated Time of Arrival |
| GMT | Generic Mapping Tools |
| TDMRC | Tsunami and Disaster Mitigation Research Center |
| GEBCO | General Bathymetric Chart of Ocean |
| SWE | shallow Water Equations |
| RANS | Reynolds-averaged Navier-Stokes |

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
