# Peer review of "Numerical Simulation of Morphological Changes due to the 2004 Tsunami Wave around Banda Aceh, Indonesia"

_geosciences, doi:10.3390/geosciences9030125_

Round 1

Reviewer 1 Report

Review: Rasyif et al., Numerical Simulation of Morphological Changes due to the 2004 Tsunami Wave around Banda Aceh City, North Sumatra, Indonesia

This paper describes the application of a combined hydrodynamic and sediment transport model to investigate changes to coastal morphology around Banda Aceh during the 2004 Indian Ocean Tsunami. The results show that the model is able to reproduce the gross features of coastal morphological change as observed in pre- and post-event satellite imagery. In general, the paper is well-written and concise, and covers a topic of interest relevance to Geosciences. The following revisions are suggested to improve the manuscript.

My main suggestions for improving the manuscript are:

-       The paper focuses a lot on the observed coastal changes and the ability of the model to reproduce the large scale features of this change. However the coastal morphology change itself has been documented previously (Griffin et al. 2013) as have previous attempts to model this change for nearby parts of the Aceh coast (Syamsidik et al 2013). I think the paper could be strengthened by two further aspects:

1.     Analysis of the model’s sensitivity to the data (e.g. topography and sediment grainsize) used in implementing it; and

2.     A more robust discussion of what we can learn about the dynamics of coastal morphology change during large tsunami. While the coastal changes have already been observed and documented previously, what this study could really add is some insights into the processes that lead to this change. I would expect some sensitivity analysis, exploring different model setups, would allow for some insights to be gained.

-       The introduction could be strengthened through greater reference to the literature. As mentioned above, Griffin et al. 2013 analysed coastal geomorphology changes in Aceh using satellite imagery (in a study very similar to the analysis shown in Figure 10) and should be cited here. However, what should be emphasised in this study is that you are trying to model the processes leading to this coastal change, rather than just observing these changes. There have also been a large number of studies of coastal change following the 2011 Tohoku tsunami – this is mentioned in passing, but the study could be strengthened by sitting itself more within this literature. These include Richmond et al. 2012; Tappin et al. 2012, Tanaka et al. 2012. Probably there are others and the authors should check this for comparison with the methods and results of this study.

-       The initial condition for coastal topography is not indicated, only a reference to GEBCO, which is quite coarse. Surely higher resolution data was used? How good is the pre-2004 topography data? Does it capture the barrier systems and coastal lagoons well? This would be expected to have a significant impact on the results.

-       A constant grain-size is applied to the entire study area. Some analysis of the potential errors due to this assumption is needed. How does the median grainsize of sand barriers really vary compared with the back-barrier lagoons and fish farm areas? Are you assuming that all sediment is transported from offshore or the beach onshore, covering presumably finer grained environments behind the barriers? How sensitive are your results if you change the median grainsize? How does your method compare with those that model a mixed grainsize distribution, such as Gusman et al. (2018).

-       Section 5.3 (Coastal profile changes) is quite weak. As far as I can tell, the pre-tsunami coastal profile is only an estimate, not directly take from high resolution elevation data. And then the post-event measured profile is not immediately after the tsunami. Perhaps this could be strengthened by indicating, based on pre-event imagery, which parts of the coast were sub-aerial and which where below sea level.

-       Related to above, the statement at L217 that the bathymetry data has changed between 2004-2006 cannot be made based on the evidence shown here. It could also just be that your model is incorrect (possibly because the initial condition is not well known).

-       In comparison with Section 5.3, Section 5.4 is quite strong as you have good pre- and post-event data to compare your results with. I would emphasise this component of the paper as validation of your method. 

-       Are you able to comment further on where the eroded sediment in your model is deposited as discussed briefly in L202-205? In particular, are there studies of post-event sediment deposits that could be used to further validate the model? E.g. studies that show barrier-derived sediments were deposited in the ponds and lagoon areas as indicated in the model results? This kind of validation would really strengthen the paper, if the data is available.

-       A rephrasing of the conclusions. E.g. Statements such as ‘This study indicated that a tsunami wave is capable of destroying the land barrier.’ (L265) are incorrect. We already know that this is the case from the observations (e.g. satellite imagery), and as mentioned above this has been reported previously for Aceh. What should be emphasised more here is that you are able to model these changes (this is emphasised more the 1st paragraph of the conclusions).

-       Related to the above comment, can the modelling provide greater insights into how the coastal changes occurred? Were multiple waves required?

Minor comments

Title: Suggest changing North Sumatra to either ‘northern Sumatra’ or ‘Aceh’. As Banda Aceh is not located in the province of North Sumatra (Sumatra Utara) this could be confusing.

L7: reveals

L9: ‘proven’ seems too strong a word, perhaps say ‘shown’

L10: What is meant by ‘non-linear’ effects? This isn’t really discussed in the rest of the paper, so doesn’t need to be mentioned here.

L36: Put abbreviations directly after the phrase they stand for, rather than repeating it.

L41: It’s unclear what is meant by ‘short-term’ changes here, and how this differs from the current study, which also looks at short term changes.

L67-69: The sentence about Mw > 7 earthquakes is irrelevant and should be removed.

L79: Remove ‘were’

L80-81: Is there a reference for this statement?

L85-86: Can a reference or link to where the model is available be provided here?

L88: Should read Lhok Nga, Aceh and Thailand for readers not familiar with Lhok Nga’s location.

Section 3.2 The sediment transport model is characterised by parameters where a range and default values are given (i.e. fts, Ts,min); however, the actual values used in the model set-up need to be given.

L162: Initial condition of topography is not given. A statement regarding accuracy for pre-2004 topography is needed.

Figure 6: I’m not sure this is needed. This is a coastal change study, not an inundation study, so the latter figures (in particular 10 and 11) are more important.

L220: 61600 m. Is this an error? And where is this distance measured from?

Figure 10: Add dates and source of imagery to the caption

References:

Griffin, C., Ellis, D., Beavis, S. and Zoleta-Nantes, D., 2013. Coastal resources, livelihoods and the 2004 Indian Ocean tsunami in Aceh, Indonesia. Ocean & coastal management71, pp.176-186.

Gusman, A.R., Goto, T., Satake, K., Takahashi, T. and Ishibe, T., 2018. Sediment transport modeling of multiple grain sizes for the 2011 Tohoku tsunami on a steep coastal valley of Numanohama, northeast Japan. Marine Geology405, pp.77-91.

Richmond, B., Szczuciński, W., Chagué-Goff, C., Goto, K., Sugawara, D., Witter, R., Tappin, D.R., Jaffe, B., Fujino, S., Nishimura, Y. and Goff, J., 2012. Erosion, deposition and landscape change on the Sendai coastal plain, Japan, resulting from the March 11, 2011 Tohoku-oki tsunami. Sedimentary Geology282, pp.27-39.

Tappin, D.R., Evans, H.M., Jordan, C.J., Richmond, B., Sugawara, D. and Goto, K., 2012. Coastal changes in the Sendai area from the impact of the 2011 Tōhoku-oki tsunami: Interpretations of time series satellite images, helicopter-borne video footage and field observations. Sedimentary Geology282, pp.151-174.

Tanaka, H., Tinh, N.X., Umeda, M., Hirao, R., Pradjoko, E., Mano, A. and Udo, K., 2012. Coastal and estuarine morphology changes induced by the 2011 Great East Japan Earthquake Tsunami. Coastal Engineering Journal54(01), p.1250010.

Author Response

We appreciate you taking the time to offer us your comments and insights related to the manuscript. We found your feedback very constructive. Revision based on the reviewer suggestion is shown using red highlight in the manuscript.

Reviewer 2 Report

Please see attached comments

Author Response

Thank you for taking the time and energy to help us improve the paper, we really appreciate all your insightful comments. We worked hard to revise the manuscript based on each of the reviewer comment. Revision based on the reviewers suggestion is shown using red highlight in the manuscript.

Reviewer 3 Report

This paper presents the results of numerical simulation of tsunami inundation and sediment transport as applied to the impacts of the 2004 Indian Ocean tsunami on the coastal area of Banda Aceh, Indonesia.  The model results are compared to bathymetric data along three cross-sections in the coastal zone as well as satellite data that show morphologic changes in map view.  The model predictions compare favorably with the actual data.  These results are significant because they show such modeling can be an effective tool for land management and for predicting areas that are likely to be effected by erosion or deposition due to tsunami inundation. The paper is fairly well-written and well organized, but needs minor revision to correct language usage errors and to clarify sentences in a few areas as indicated in the annotated manuscript.

Author Response

We greatly appreciate the reviewer’s efforts to carefully review the paper and the valuable suggestions offered. In the following pages, we addressed responses to each of the comments of the reviewer. Revision based on the reviewer suggestion is shown using red highlight in the manuscript.

Round 2

Reviewer 1 Report

The updated version of the paper address a number of important points in the original review and the authors have clearly put in a good effort. However, the most significant comments from my previous review still stand:

- The topography and sediment (grainsize) data from before the 2004 Indian Ocean Tsunami is poorly known, and a number of default parameters are used in the model. The authors have been more explicit in the paper about what data was used and this is a good improvement. However, I still feel that they do not comment enough on what impact this may have on the results - and therefore the overall conclusions of the paper are potentially compromised. In short, given the uncertainties in input data I am not yet convinced that the good agreement between the model results demonstrates good model performance and isn't just by chance. I totally understand that the authors can only work with the data they have, but given the data limitations they really need to do some more work to understand/explain what impact this may have on their results. I previously recommended some kind of sensitivity analysis, varying the unknown parameters to see if this significantly changes the results. In some cases, the authors may also be able to point to relevant literature that can give a qualitative indication of the kinds of uncertainties that could be expected. E.g. Griffin et al 2015 showed how topographic uncertainty could significantly impact modelled inundation extent. I am not aware of similar literature dealing with uncertainties in sediment transport model input data, but I would recommend the authors do a search on this. 

I feel that the above is a major point and should be addressed. Aside from that, there a still a few English grammatical errors throughout the paper and I would recommend another careful proof-read. 

Reference: Griffin, J., Latief, H., Kongko, W., Harig, S., Horspool, N., Hanung, R., Rojali, A., Maher, N., Fuchs, A., Hossen, J. and Upi, S., 2015. An evaluation of onshore digital elevation models for modeling tsunami inundation zones. Frontiers in Earth Science3, p.32.

Author Response

We greatly appreciate the reviewer’s efforts to carefully review the paper and the valuable suggestions offered. We found your feedback very constructive. We hope that the revisions can meet the journal publication requirements.

Reviewer 2 Report

The manuscript has been significantly improved. 

The new figures make the manuscript much easier to interpret. I still believe a slightly more balanced view on the model performance is warranted. In cases where there is discrepancy the authors dismish this with e.g. morphological changes in the intermediate period. It might also be a case of the model not capturing the physics. Furthermore, I understand the choice of  a single grain size and making the whole area erodible, but I think that some consideration to the possible impact of this should be made in the manuscript.

While the manuscript has been improved, a do not have the same quality in writing as the originally submitted manuscript (please see comments in attached annotated manuscript).

If a slightly more balance view on the model performance is made and changes suggested in the annotated manuscript are considered, I will recommend publication.

Author Response

Thank you for taking the time and energy to help us improve the paper, we really appreciate all your insightful comments. We worked hard to revise the manuscript by addressed each comment. We hope that the revisions can meet the journal publication requirements.

Round 3

Reviewer 1 Report

The authors have done a good job of dealing with the previous review comments. I feel that the paper now more clearly acknowledges the limitations of the model's outputs due to limitations in the input data and model parameterisations, and therefore the conclusions drawn are better justified.

Minor point: L125 Suggest remove 'where'.